# Interventions to Foster Resilience in Family Caregivers of People with Alzheimer’s Disease: A Scoping Review

**DOI:** 10.3390/ijerph21040485

**Published:** 2024-04-16

**Authors:** Lucía Santonja-Ayuso, Silvia Corchón-Arreche, Mari Carmen Portillo

**Affiliations:** 1Paterna Mental Health Unit, 46980 Valencia, Spain; santonja_luc@gva.es; 2Faculty of Nursing and Chirody, University of Valencia, 46010 Valencia, Spain; 3NIHR Applied Research Collaboration Wessex, School of Health Sciences, University of Southampton, Southampton SO171BJ, UK; m.c.portillo-vega@soton.ac.uk

**Keywords:** Alzheimer’s disease, carer, family caregiver, psychological resilience, spouse caregiver

## Abstract

The family caregiver of a person with Alzheimer’s disease still experiences, in most cases, negative consequences in their biopsychosocial environment, which are related to the acquisition of this role. However, it has been observed that this fact is not universal in this type of population since benefits can be obtained in the act of caring through the development of resilience. Given this possibility and given that nurses are the health professionals who support people in this illness process, there is an urgent need to identify which non-pharmacological interventions could improve or promote resilience in family caregivers of people with Alzheimer’s disease. Therefore, our overall objective was to determine which interventions are useful in promoting resilience in family caregivers of people with Alzheimer’s disease through a scoping review. The data were analysed using an adapted version of Arksey and O’Malley’s methodological framework, after critically reading the articles with the CasP and MMAT tools. Nine articles were included (five analytical experimental, two quantitative and two mixed). Three types of interventions related to promoting resilience in family caregivers of people with Alzheimer’s disease were identified: meditation, multicomponent psychoeducation and creative art; nurses participated as co-therapists in the last two.

## 1. Introduction

According to the World Health Organisation [1], the population has undergone an unprecedented demographic change due to the scientific and socioeconomic development made in the last 50 years, during which there has been an increase in life expectancy and old age survival. Consequently, age-related chronic pathologies have become particularly relevant in society, as populational morbi-mortality and alterations related to functionality and independence have increased. In fact, dementias (and especially Alzheimer’s-type dementia) should be noted in this group, since they are the main age-related chronic disease that cause disability and death, particularly from the age of 65 [2,3].

It is estimated that a total of 131.5 million people worldwide will present an Alzheimer’s-type dementia by 2050 [4]. Alzheimer’s disease is known for being a chronic neurodegenerative disease (of unknown cause) whose onset is insidious and affects superior functions like memory, speech or orientation, which cause a significant behaviour alteration [5]. Therefore, the estimated costs related to caregiving for this type of pathology in the year 2050 are expected to exceed EUR one trillion [6]. Among them, costs that stand out the most are the ones invested in long-term caregiving, and specifically the ones related to the habituation of the place of residence of the person with Alzheimer’s disease and informal caregiving [7].

Informal caregivers are usually women over 40 and even 65 years of age, who take care of their husbands, who are over the age of 70 and have any stage of Alzheimer’s disease. They take care of their basic needs without any economic compensation. These needs are harder to satisfy as the disease progresses and they can cause a negative symptomatology that directly affects the health of the caregiver as a result of caregiving [8,9]. This symptomatology is called the ‘caregiver syndrome’, which is more intense in caregivers of people with Alzheimer’s disease due to its features [10,11]. However, some studies state that this syndrome does not have to appear in all family caregivers, in fact, it is even possible to obtain benefits from the experience of caregiving through the development of resilience [10,12].

Resilience, understood as the ability to effectively adapt to a significant adverse environment while learning and developing skills that generate positive experiences and emotions, is a dynamic concept that can be learned. It helps caregivers to avoid the experience of personal, social or health-related failure [10,12]. The evidence proves that people who are more resilient present a better psychological adaptation to the behavioural changes that come with the disease, a greater development of new coping strategies in response to care demands and, in general, a better welfare state, since levels of anxiety, depression, psychotropic drug use or external resources use (such as social health or institutionalisation centres) are reduced [13,14].

As resilience can bring benefits at a general level, it is increasingly present in the health field. In fact, three nursing diagnoses related to ‘risk, deterioration and willingness to improve resilience’ were included in 2009 in the nursing field, which held nursing professionals responsible for their care in the general population [15]. Considering that nursing professionals are one of the health professional groups who support caregiving dyads (informal caregiver—person with Alzheimer’s disease) through the disease process and with their health as whole, there is an urgent need for developing action plans to promote resilience in this type of population, given the bio-psycho-social advantages that it presents. So far, most studies have been focused on reducing negative symptomatology (anxiety, stress and depression) [16,17].

Although the first classification of interventions for the improvement of the welfare state of family caregivers of people with Alzheimer’s disease (which includes counselling and psychoeducational, multicomponent and breathing and resting interventions) was carried out by Bourgeois, none of them have considered resilience as such [18]. Furthermore, in the general scientific literature, resilience has been studied mainly through descriptive studies on other types of populations (children, oncological or palliative patients, people in grief) or with very variable methodological designs [19,20,21]. Consequently, a very small number of studies has been able to develop positive skills like resilience in family caregivers of people with Alzheimer’s disease from a more positive and salutogenic perspective.

Hence, our research work aims to answer, through a scoping review, the next research question: “Which non-pharmacological interventions seem promising in order to foster resilience in informal caregivers of people with Alzheimer-type dementia?” We also aim, as a general objective, to find out which interventions have been useful in order to foster resilience in family caregivers of people with Alzheimer’s disease; and, as specific objectives, we aim to: (1) to understand and compare the concept of resilience in the studied population; (2) identify health outcomes that have been studied together with resilience; (3) find out the profile of the family caregivers of people with Alzheimer’s disease; (4) find out nursing professionals’ implication in the development of the studied interventions; (5) describe the main features of the interventions to approach resilience.

## 2. Materials and Methods

### 2.1. Design

Since resilience is a broad concept that has been poorly explored through randomised clinical trials, whose outcomes may be heterogeneous in terms of interventions or the type of population [22], a scoping review, in which we followed the methodological framework of Arksey and O’Malley [23], was chosen to answer the objectives. This methodological framework is based on: (1) identifying the research question; (2) identifying relevant studies; (3) selecting studies; (4) charting data; (5) compiling and summarising the obtained outcomes.

### 2.2. Research Question

Following the initial question, three others were developed to help conduct the review: (1) What is the profile of the main caregiver to whom these interventions are applied? (2) What other results have been studied in the healthcare field together with resilience? (3) What is the role of nurses in this type of interventions?

### 2.3. Search Strategy

An initial search of the Web of Science database was carried out following the PICO research question, from which the third component was excluded, as shown in Table 1 [24]. These terms referenced the type of population (person with Alzheimer’s disease and their family caregiver), interventions and results that can be achieved (resilience) and were obtained by reading relevant articles on the subject, identifying synonyms and consulting MeSH terms like ‘Alzheimer disease’, ‘family caregiver’ or ‘psychological resilience’. The terms of each group were combined with each other with Boolean operators OR and AND, and were adapted to the specific search of each database from 2021 to the last quarter of 2022. The following databases were chosen because they were the most representative ones in the social healthcare field [25]: Medline, CINAHL Plus, PsycINFO and Cochrane Library.

The year of publication of the articles was taken into account because resilience was included as a nursing diagnosis and MeSH term in 2009. Language limits were also applied (research in English and Spanish, in order to accommodate the researcher’s language skills), as well as age of the caregivers (at least 18 years of age) and age of people diagnosed with Alzheimer’s disease (at least 65 years of age, as age is the main determining factor of prevalence of this disease and its aetiology) [2,26]. With the latter, the aim was to exclude studies where participants presented early Alzheimer’s or those in which dementia was related to other types of chronic neurological alterations like brain tumours or Parkinson’s disease, among others. During the final phase of the process, a manual search of the most relevant publications was conducted to assure their inclusion in case of meeting the established inclusion and exclusion criteria.

The initial search strategy was conducted following the recommendations of the PRISMA guidelines adapted to scoping reviews (PRISMA-ScR) [27].

#### Results Management

Bibliographic software Mendeley (version 2.98.0) was used for administering references, searching for and deleting duplicates [28].

### 2.4. Selecting Studies: Inclusion and Exclusion Criteria

The inclusion and exclusion criteria established for determining which articles were selected for the scoping review are indicated in Table 2.

### 2.5. Selection of Evidence Sources

After the removal of duplicates, a total of 156 articles were fetched from the searches of the targeted databases. Having read the titles and abstracts, 73 articles were retrieved for full text reading according to the established inclusion and exclusion criteria.

Eventually, after having carried out a critical reading of 12 articles, a total of 5 articles were obtained, along with 4 others that were found through manual searches.

Figure 1 shows a detailed description of the study selection, which notes a total inclusion of 9 articles used for the elaboration of the scoping review.

#### 2.5.1. Quality Appraisal

According to the article’s methodological design, CASP guidelines [29] were used for the assessment of the methodological quality of qualitative and analytical studies. The MMAT guideline was used in studies with mixed methodology [30]. To achieve this, an Excel spreadsheet with the items to be assessed from each article was made. The ones that did not meet the minimum criteria established by the guidelines were excluded (Appendix A).

#### 2.5.2. Outcomes of the Methodological Critique of the Articles

The methodological assessment of the articles allowed us to identify that research carried out in this field was mostly analytical-experimental (*n* = 5) [31,32,33,34,35], followed by qualitative-design articles (*n* = 2) [36,37] and two mixed-type articles [38,39].

Although randomised clinical trials presented a clear and well-defined research question, none of them complied with the items recommended by CaSP. These studies show scarcity in the intervention’s magnitude effect, its own benefits, its implementability and the positive benefit–risk balance. Consequently, the obtained results were questioned, and it was evidenced that there was a need for carrying out more homogeneous studies with higher scientific quality. Or, on the contrary, the need to combine randomised clinical trials and qualitative research, as resilience is a complex intrapersonal concept [19], could potentially be considered.

It is worth noting that just one of the qualitative studies fulfilled the CaSP criteria. Despite this, both qualitative articles were taken into account for their resilience-related positive outcomes, as well as for their implementability in the surrounding context of the caregiver. This fact reinforced the indication that mixed studies may need to be carried out and resilience may need to be taken into account on both a quantitative and qualitative level, since it is a psycho-social concept that can directly and indirectly influence the caregiver’s integral sphere.

Lastly, none of the mixed methodology studies met the established MMAT criteria to be considered high-quality. Despite this, they were included as part of the review, since the main objective of both articles was inherently connected to resilience and intended to obtain more complex and enriching outcomes from both perspectives.

### 2.6. Data Charting and Synthesis

Given the variability in the research designs in the articles we found, the recommendations of Arksey and O’Malley [23] for the extraction of data designed for scoping reviews were followed, which were adapted and are projected in Table 3. However, despite having looked for a uniform approach for data grouping, the required information could not be retrieved because the selected articles lacked relevant material.

Data were descriptively and narratively analysed according to our objectives and considering the nature and distribution of the studies included in the review.

Firstly, attention was paid to the studies’ design and objectives and a variety of interventions included was in the review. An attempt was made to perform a categorisation similar to the one of Bourgeois et al. [18], but it was not possible upon the presentation of two new types of intervention.

Secondly, specific features were searched for in each study, paying attention to the concept of resilience and its studied variables; the programme’s target group; particularities of each intervention (given that variabilities that had to be considered were found within the same group); and what type of professionals carried out the interventions.

Comparisons were attempted to be made if two or more studies were sufficiently homogeneous in terms of the type of intervention and outcomes. In the event that there was no possibility of performing them, an attempt was made to highlight common elements or those which were contradictory.

## 3. Results

A total of nine articles were eventually included in the review, of which three were conducted in the UK, two in the US and one in Australia, Iran, Colombia and Asia. Among them, five were analytical experimental (three randomised clinical trials and two quasi-experimental studies), two of them were of mixed methodology, and two were qualitative.

All of them studied resilience building exclusively in the family caregiver, except for two of them, which studied it in the dyads [34,39].

Since resilience was part of the general and specific objectives of the included studies, and according to the general features of the interventions found, a general classification of them was established: interventions based on psychoeducation, creative art and meditation.

### 3.1. Resilience

The understanding of the concept of resilience varied among the studies. The concept was not specifically defined in four of them [33,35,37,39], but the definition that indicated that resilience consisted of overcoming obstacles or positive adaptation to adversity was emphasised in three of them [32,36,38]. The Wagnid and Young resilience scale was again used for the assessment of resilience through measurement tools [31,34,35,38], although the use of other less specific but currently not validated scales was noted [33,39].

After the performance of intervention programmes, an invariability [33,34], an increase [31,32,35] and a decrease [38,39] in the resilience values were found. According to the authors, this fact could be related to the sample composition (it was small and was mainly composed of women, who generally scored worse), to the programme implementation in the caregiving dyad, and to the scarce interest people had in the activity or a lack of adherence to the intervention.

### 3.2. Other Health Outcomes That Have Been Studied Together with Resilience

Resilience has been studied together with other types of variables. The ones that were part of the negative symptomatology derived from care were most prevalent, for example: depression [31,32,33,34,38], overload, stress [33] and anxiety [32].

When trying to look for certain congruence with these types of variables and outcomes in resilience, it was observed in some cases that when resilience levels increased, negative variables decreased [32,35,39] or remained the same [33]. In the latter case, the authors state that it happened because of the performance of the programme in the caregiving dyad and in the female population or because of a lack of adherence to the intervention.

As for positive variables, these included quality of life [33,34,39] and mental health [31,32,33,34,35,38,39]. Since they were single studies, no significant conclusions could be extracted, given that, in some cases, these variables remained the same [33], increased [31,32,34,35] or decreased [38].

Both qualitative studies, which were based on creative art interventions, coincided with a subjective decrease in stress and feelings of guilt [37]. Family caregivers also experienced an improvement in their mental health [36,37], concurrently with resilience.

### 3.3. Profile of the Caregiver Receiving the Intervention

Participants in the study were mainly women (between 41 and 80 years of age) who were married to a husband diagnosed with Alzheimer’s disease, were cohabitants and also took care of the household chores [31,33,34,35,36,37,38,39].

The only study that differed in this aspect was that of Ghaffari et al. [32], as its sample was represented by daughters (with an average age of 43 years) cohabiting with the person diagnosed with Alzheimer’s disease. There have been several authors that have given importance to this sociodemographic variable and have stated that resilience in family caregivers is greater in countries culturally embedded in spirituality [32,35], which increases with the age of the caregiver [32,35,38].

However, other variables caused controversy: inversely proportional relations between resilience and sex (being a woman) were found in some studies [35,38], but not in others [32]; outcomes directly proportional to kinship (being married), social status (upper middle class) or educational level were also found [35], as opposed to the study of Ghaffari et al. [32], which could not prove some of these associations.

### 3.4. Implication of Nursing Professionals and Main Features of the Interventions

#### 3.4.1. Nursing Professionals

Nursing professionals participated as therapists in a creative art intervention [38] and in a multicomponent psychoeducation intervention [34], as observed in Table 3.

Nevertheless, the role they performed as part of the healthcare team and the activities to develop within the programme were unknown.

#### 3.4.2. Interventions

##### Interventions Based on Creative Art

Creative art interventions are activities that use different means like theatre, music, dance and literature in a creative way, with the aim of improving the physical and mental health of people, allowing stimulation and development of new expression skills [40].

Four articles applied the creative art interventions [36,37,38,39]: two of them were qualitative (through grounded theory) and two of them were mixed, in a total of 20 caregivers and 32 dyads.

Within this group, the performed activities were: arts and crafts [37], painting [36], poem writing [38] and theatre staging [39].

The role of creative arts as a method to foster resilience is inconclusive because both mixed studies [38,39] did not obtain significant positive outcomes for resilience, and one of them presented an increase in depression and overload near to the end of the programme. Caregivers had a feeling of helplessness post-intervention [38].

Although both of them were carried out over 8 weeks, the case of Kidd et al. [38] was performed in the participants’ places of residence and on an exclusive sample of caregivers, who had to write a poem every week, and the study of McManus et al. [39] applied performing arts in the caregiving dyad, in which it was highlighted caregivers need to obtain skills for the management of the disease, the addition of other psychoeducative activities and a reduction in the onset of negative events (like stress), while performing the activity.

When it comes to the qualitative studies (carried out exclusively with a total of 10 family caregivers), activities like painting (in the place of residence) or going to art galleries, together with other types of arts and crafts performed over 5 weeks, had positive effects on resilience and mental health. They made it possible for caregivers to have a feeling of ‘rest and relief’, and allowed them to better process changes in Alzheimer’s disease; in fact, after the completion of this regime, the group continued it, as it was more useful for them than the self-help groups they used to attend [36,37].

##### Multicomponent Interventions Based on Psychoeducation

Multicomponent psychoeducative interventions seek to develop coping skills, to adapt to the new role through psychological support and to acquire specific knowledge related to the pathology [41].

This category includes three studies: two of experimental design [32,34] and one of quasi-experimental design [31], with a total of 62 family caregivers and 365 dyads. Cognitive behavioural [31,34] and resilience-oriented psychoeducation approaches [32] were in these studies.

The first two studies presented different outcomes when applying cognitive behavioural therapy to caregivers and dyads, since the latter resulted in no improvement in the neurocognitive ability of people with Alzheimer’s and no modifications regarding the caregiver’s general welfare state, which slightly increased depression values upon the completion of the study [34]. Authors stated that implementing cognitive behavioural rehabilitation in a chronic neurodegenerative pathology where the caregiver had to be trained to conduct the programme at the place of residence was not feasible to improve the quality of life of the caregiving dyad. However, the performance of these activities altogether could improve their relationships.

Positive outcomes were found in all variables of the study that exclusively focused on resilience and family caregivers because it was an 8 week, on-site, psychoeducative resilience programme. Resilience outcomes were similar in the case of Cerquera et al. [31], which also had an on-site format, although conclusions about the reduction in variables like depression and overload could not be extracted. This can be explained with the following concerns: high abandonment rate of participants due to unavailability, displacement necessity, lack of motivation for and implication in the activity and negative perspective of the caregivers as the disease progresses.

Data concerning adverse events derived from the programme are unknown.

##### Interventions Based on Meditation

Meditation entails a variety of practices (of spiritual or contemplative type) that involve the mind and body, while other activities like mindfulness (understood as the awareness resulting from the act of paying attention to sensations of the present moment without judgement) are more of a reflective or awareness-raising type of practice [42].

Both articles included in this section were analytical experimental: one of them was carried out through a randomised clinical trial in which transcendental meditation was used, in Australia [33], while the other one was quasi-experimental and was carried out through mindfulness, in Asia [35]. These interventions were carried out for a total of 202 female family caregivers who were married to the patient diagnosed with Alzheimer’s.

The outcomes between studies were contradictory: the study of Pandya [35] presented an improvement in the resilience outcomes and had lower rates of abandonment. Its sample size was bigger and it was a mixed-type study (mindfulness exercises on-site and at the place of residence). Despite having implemented it over 5 years, no adverse effects were described and it was stated that the outcomes were associated with attending at least 75% of the programme’s sessions.

The study of Leach et al. [33] revealed a decrease in stress and invariability in the resilience score, in which the onset of severe negative effects (paraesthesia, headache, blurred vision) and mild–moderate effects (arthralgia) derived from the intervention were highlighted.

## 4. Discussion

As far as is known, this is the first scoping review that identifies the most useful non-pharmacological interventions for improving and maintaining resilience in family caregivers of people with Alzheimer’s disease, who cohabit and reside in the community. In addition, it is also the first review that focuses on intervention content and techniques used in qualitative, quantitative and mixed designs.

This review identifies which activities allow for resilience building in this type of population and highlights the scarcity of interventional studies, although the concept of resilience was adapted to the Health Sciences from the 1970s and was included as a nursing diagnosis more than a decade ago. Methodological design, programme development and concrete involvement of nursing professionals are also scarce.

In relation to the general objective and taking into account the narrative analysis of the contents and techniques used, this review has various practical implications for the use and design of interventions that foster resilience in the family caregiver of a person with Alzheimer’s disease: psychoeducation, creative art and meditation activities. It is an innovative categorisation that has made it possible to broaden resilience-related intervention approaches, which have been framed within general individual or group psychological therapy, pharmacological treatment and rest resources [13,18,43].

It has been observed that the concept of resilience was defined and quantified differently in each scientific study. Although the scientific literature already indicated that it was an abstract concept which was hard to define [12,44,45], these conceptual differences have allowed for the existence of confusion biases, especially in studies in which the concept of resilience has not been clearly and concisely defined [33,35,37,39].

We consider that future resilience research should opt for the normalisation of the concept, whose main definition could concern the proper adaptation of a person when facing an adverse situation and acquisition of positive experiences [15,46]. This would allow for the application of validated scales like those of Wagnild and Young [47] and Connor-Davidson [48], and the achievement of harmonisation and consistency of processed data.

It has been interesting to learn that an improvement in resilience can be obtained directly and indirectly through the reduction of other negative variables like overload or depression. Although most studies were focused on reducing the negative symptomatology derived from caregiving, scientific evidence states the need for studying the variables of ‘overload’ and ‘depression’ in caregivers jointly, as they are the most intense and dysfunctional variables [31,49,50]. However, our findings are not consistent with these studies, since only one of them fulfilled this premise [38]. Even though the pathogenic perspective (reduction of negative symptomatology) still prevails with the possibility of increasing resilience through the improvement of other positive variables like humour, sense of identity and confrontation, future research should be based on a more salutogenic perspective in order to develop more resilient self-care skills.

As for experimental studies, it was concluded that data must be carefully analysed due to the heterogeneity of the sample sizes, variables of the studies and development of the interventions. In fact, it can be observed that some of the studies had positive resilience outcomes [31,32,35], invariable outcomes [33,34] or a post-intervention reduction in the positive outcomes [38,39]. The main reasons why positive outcomes could not be extracted and that future research should take into account are: the appropriate choice of the main activity, since participants needed both ludic and informative or demonstrative activities regarding disease management [39]; the impossibility of continuing the activities after the end of the research [38]; the low motivation and participation of the sample, due to multiple factors related to disease progression (increase of care demands, death of the person with Alzheimer’s disease) [31,32,33,34,35,38,39] and activity application in the dyad [34,39]. The latter fact coincides with other research, in which it was advised to perform the intervention exclusively with the caregiver, since resilience is a dynamic concept that can be learned (although it requires the caregiver to have a good neurocognitive ability) [3,10,12].

Regarding the profile of the main caregiver, the female gender still predominates as being responsible for the main care of people with Alzheimer’s disease. Caregivers are generally women, older than 40 to 60 years of age, who are not only in charge of the needs of the person with Alzheimer’s disease, but who also take care of household chores [51]. However, more homogeneous studies involving male caregivers should be carried out, given their increase in the last few years, to assess whether these interventions could be useful on a general level [52].

Aside from gender, this review has found conflicting results among other types of variables that have turned out to be risk or resilience protection factors, for example, age (being older versus being young) [32,38], advanced dementia stage, time spent in caregiving, level of studies and economy or drug consumption [32,35,38]. We added culture and religion as protective factors, as did the study of Pessotti et al. [53], which states that the most religious caregivers present lower levels of depression and indirectly foster their resilience levels. As for kinship, being a wife was presented as a risk factor in our review, contrary to other studies that state that it was a protective factor [54].

If the particularities of the intervention are taken into account, there is not enough evidence to confirm that the performance of activities at the place of residence is more appropriate. Nevertheless, it is true that performing them on-site involved pretexts related to abandonment of the programme like mobility problems, activity hours or the need to find another caregiver [31,32,39]. Therefore, the literature recommends activities to be performed at the place of residence or a location close to this, with enough materials and personal resources to accommodate the person with Alzheimer’s disease [34,37,55].

Studies stated that proper supervision and tracking by professionals should be ensured in case of activities being performed at the place of residence, although this was not the case in most of them [34,38]. Information and communication technologies could be used for this purpose, since they would provide necessary support and allow for better adaptation to the context and schedules of caregivers, acting as a positive reinforcement for adherence to the treatment [34,56].

In both cases, and even though the duration of an activity would be an important factor to take into account [32,33,34,35,38,39], it would be more convenient and beneficial for participants to be able to implement the previously guided intervention on their own without the need for professional supervision, especially in the application of psychoeducation and meditation activities [32,35], which seem to have obtained the most promising results.

The participation of nursing professionals was observed in two out of nine of the interventions [34,38], taking into account that two of the studies did not specify the type of participating professional [31,32]. These interventions are based on psychoeducation (with the help of psychologists and occupational therapists) and creative art.

None of them specified the role that the nursing professional played within these programmes, so it remained open to interpretation. Since nurses can participate in any of these activities, even in mindfulness activities (regardless of what was found in our review) [57], a programme in which the nursing activities are defined and assessable is required.

As part of the multidisciplinary team to which they belong, nursing professionals are one of the main figures that support caregiving dyads with their health and throughout the disease process, even before a formal diagnosis is made [17]. This fact means it could be possible to perform these activities in the earliest stages of Alzheimer’s disease, in which caregivers usually present lower levels of overload and depression and show more interest in learning disease management [13,58], and its application could be useful when symptomatology related to caregiver syndrome is starting to be detected in any type of caregiver.

Qualitative findings must also be considered carefully because certainty in the evidence was low, which can be translated into lack of confidence in the obtained outcomes. The lack of objective evidence on the improvement of the outcomes of the variables did not necessarily mean that the applied activities were not useful; in fact, they provided a different perspective on the acquisition of resilience that caregivers had thanks to creative art activities (given that the only two qualitative studies within the review addressed this matter). Therefore, it would be interesting to develop these types of studies with an increased methodological quality, to extend them to other type of activities like psychoeducation or meditation, and/or to complement them with measurable data and objectives, with the final aim of having a better comprehension of the concept of resilience, contextualising interventions and adapting them to the needs of family caregivers.

### Strengths and Limitations

Firstly, only studies that declared to have a main or secondary objective of fostering resilience (in English and Spanish) were considered. Closely related terms to resilience like ‘quality of life’ and ‘confrontation’ were excluded. However, this fact could be considered as a strength because the concept of resilience has its own meaning. It could be confused with the terms ‘confrontation’ and/or ‘adaptation’, among others, but they are not synonyms; thus, it has been possible to obtain research that explicitly considers resilience.

Secondly, even though four of the articles included in this scoping review had a qualitative or mixed design (in which individual experiences are prioritized over analytical metrics), the number of participants was small. Given the lack of methodological rigour in most studies and their large heterogeneity presented within the same category of intervention, it has not been possible to perform an exhaustive comparison between studies. This made it impossible to draw conclusions that confirm which activity and/or element of the intervention allowed for the furtherance of resilience in family caregivers of people with Alzheimer’s disease. Designs that meet all methodological quality criteria are required to answer this question.

## 5. Conclusions

This scoping review concludes with low certainty that the interventions that can be useful for the improvement of resilience in family caregivers of people with Alzheimer’s disease are multicomponent psychoeducation, creative art and meditation. And, within them, psychoeducation and mindfulness have apparently shown promising results.

This review also emphasises the need for carrying out more research of high methodological quality in which a greater sample size, greater homogeneity between participants, a better interpretation and harmonisation of the outcomes are included. It would also be useful to carry out research composed of different methodologies (mixed studies) in order to identify the meaning of resilience and/or other resilience-related relevant elements, thereby deepening our understanding of this concept and the role of nurses in the activity implementation process, caregiver support and assessment and maintenance of outcomes.

In any case, this is the first scoping review that has found that the resilience of family caregivers of people with Alzheimer’s disease can be enhanced with the implementation of non-pharmacological interventions based on psychoeducation, creative art, and meditation, and nurses often take part in the interventions as co-therapists.

## Figures and Tables

**Figure 1 ijerph-21-00485-f001:**
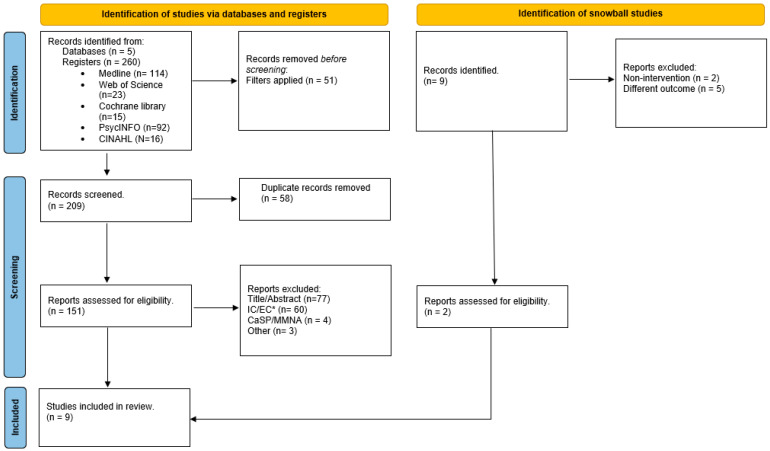
Flowchart of the PRISMA 2020 guidelines adapted to scoping reviews that included database searches, registries and other sources. IC/EC*: Inclusion and exclusion criteria.

**Table 1 ijerph-21-00485-t001:** Search strategy and search terms.

Group 1. Binding terms regarding type of chronic process.
Alzheimer Disease * OR Alzheimer’s Disease *
AND
Group 2. Binding terms regarding type of population.
Carer * OR Caregiver * OR Family caregiv * OR Spouse Caregiv * OR Informal Caregiv *
AND
Group 3. Binding terms regarding intervention.
Intervention * OR Multidisciplinary intervention * OR psychological multicomponent intervention *
AND
Group 4. Binding terms regarding outcomes.
Psychological resilience * OR Resilience *

* tool used as search operator.

**Table 2 ijerph-21-00485-t002:** Inclusion and exclusion criteria.

Inclusion Criteria	Exclusion Criteria
Studies that included people with Alzheimer’s disease in any stage of illness and/or their family caregivers.	Studies conducted on formal caregivers, understood as a person who is not part of the family, whether qualified or not to perform care, in exchange of a monetary compensation.
Studies whose population diagnosed with Alzheimer’s disease were 65 or older.	Studies in which those receiving care from family caregivers presented a neurodegenerative disease other than Alzheimer’s disease and/or were under 65 years of age.
Studies that explored or intervened on family caregivers of people with Alzheimer’s disease with a main or secondary objective of improving resilience.	Studies whose aim was not to improve resilience and/or did not used resilience as a method of eliciting other variables.
Studies involving non-pharmacological interventions.	Studies that chose external resources such as institutionalization or other types of resource or financial aid.
Full-text studies with qualitative, observational, analytical and/or mixed designs.	Letters to editorials, comments, literature reviews, guidelines, grey literature and research that did not meet quality criteria.

**Table 3 ijerph-21-00485-t003:** Data charting.

Main Author Year Country	Design Sample Main activity	Resilience Definition Measurement	Implementation Strategies	Outcomes
Pienaar L 2015 UK	Design: qualitative Sample: 4 caregivers (wives from 50 to 79 years old) Alzheimer’s disease severity: early, middle, and late stage Activity: to go to an art gallery and craft-making	Definition: No Measurement: transcription	Format: on-site (Healthy Aging Café) Total duration: 5 weeks Frequency: 4 sessions/90′ Agents: an occupational therapist and a psychologist Abandonment: 50%	Feelings of guilt and stress were reduced. Resilience and mental health took on a positive meaning.
Hunt B 2016 UK	Design: qualitative Sample: 6 caregivers (4 wives and 2 daughters from 60 to 77 years old) Alzheimer’s disease severity: middle and late stage Activity: to make a painting	Definition: Achieving relevant tasks despite adversity Measurement: transcription	Format: place of residence Total duration: Unknown Frequency: Unknown Agents: Occupational therapist Abandonment: Unknown	Psychological welfare, resilience, identity, relief and good mood were increased.
Kidd L 2011 US	Design: mixed Sample: 20 caregivers (17 women and 3 men from 41 to 80 years old) Alzheimer’s disease severity: early and middle stage Activity: poem writing	Definition: overcoming obstacles Measurement: Wagnild and Young resilience scale	Format: place of residence Total duration: 8 weeks Frequency: 1 poem/week Agents: nurses Abandonment: Unknown	Resilience and mental health were reduced. Depression and overload were increased. Conclusive quantitative outcomes cannot be drawn. Women and younger caregivers scored worse than men in all variables.
McManus K 2021 US	Design: mixed Sample: 32 dyads (wives, from 61 to 80 years old) Alzheimer’s disease severity: early and middle stage Activity: theatrical staging	Definition: No Measurement: Brief Resilience Scale	Format: on- site Total duration: 8 weeks Frequency: 1 h/week Agents: theatre instructors Abandonment: 37%	Overload and resilience caregiver were reduced. QoL* (person with Alzheimer’s disease) was reduced.
rell M 2017 UK	Design: RCT Sample: 356 dyads Alzheimer’s disease severity: early and middle stage Activity: cognitive stimulation	Definition: Synonym of QoL Measurement: Resilience Scale-14	Format: place of residence Total duration: 25 weeks Frequency: 90′/week (75 sessions) Agents: Mental health nurses, clinical psychologists, occupational therapists or research assistants Abandonment: 23%	Relationship quality (according to the person with Alzheimer’s disease) and caregiver depression were increased. QoL*,cognition in the person with Alzheimer’s disease andcaregiver’s health status remained constant.
Cerquera A 2017 Colombia	Design: quasi-experimental Sample: 30 caregivers (women, from 55 to 80 years old) Alzheimer’s disease severity: early, middle, and late stage Activity: cognitive-behavioural stimulation	Definition: personality trait Measurement: Resilience Scale	Format: on-site Total duration: Unknown Frequency: 10 sessions/90′ Agents: Unknown Abandonment: 66.6%	Mood, resilience and social support were increased. It is not possible to conclude if it is effective on depression and stress or not.
Ghaffari F 2019 Iran	Design: RCT Sample: 54 caregivers (daughters, 43 years of age) Alzheimer’s disease severity: middle and late stage Activity: resilience psychoeducation	Definition: overcoming difficulties and adaptation Measurement: Connor-Davidson Resilience Scale	Format: on-site Total duration: 8 weeks Frequency: 30 caregivers (women, from 55 to 80 years old) Agents: Healthcare professionals Abandonment: 7.4%	Mental health, age-related resilience (greater), care giving duration (lower), income level and toxic habits (smoking) were increased. Depression, anxiety and insomnia were reduced. There were no relations between gender, kinship, civil status or educational level.
Leach MJ 2015 Australia	Design: RCT Sample: 17 caregivers (wives, 66 years old on average) Alzheimer’s disease severity: middle and late stage Activity: Transcendental meditation	Definition: NO Measurement: WebNeuro scores	Format: on-site Total duration: 12 weeks Frequency: Unknown Agents: Transcendental meditation instructor Abandonment: 52.7%	QoL*, depression and resilience remained constant. Stress was reduced.
Pandya S 2019 Asia	Design: quasi-experimental Sample: 185 caregivers (Hindu wives) Alzheimer’s disease severity: early, middle, and late stage Activity: Mindfulness meditation	Definition: No Measurement: resilience scale	Format: combined Total duration: 5 years Frequency: 1 session/45′ per week Agents: Transcendental meditation instructor Abandonment: 21.6%	Overload was reduced. Resilience and auto-efficacy were increased. Gender, kinship, religion, social status, education, employment, and assistance to 75% of sessions were decisive factors.

QoL*, Quality of Life.

## Data Availability

Available data are included in the article.

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
