# Peer review of "Interventions to Foster Resilience in Family Caregivers of People with Alzheimer’s Disease: A Scoping Review"

_ijerph, 2024, doi:10.3390/ijerph21040485_

Round 1

Reviewer 1 Report

Comments and Suggestions for Authors

Interventions to foster resilience in family caregivers of people with Alzheimer disease: a scoping review

This scoping review examines the topic of resilience in caregivers of patients with Alzheimer's disease. The authors followed a correct methodology, following the most accredited guidelines. Some minor points may improve the utility of the study.

1.       Abstract. According to editorial rules, it must be a single paragraph of about 200 words maximum, without headings.

2.       The synthetic concepts reported after the abstract (“What does this paper contribute to the wider global clinical community?”) must be developed in the Discussion and reported in the Conclusions.

3.       The interventions that the authors found are very heterogeneous, as they correctly recognize. An aspect that deserves to be underlined is that the experiences refer to very small samples. This inevitably limits the reliability of the results. Authors may report this issue to readers.

4.       An element of interest to readers could be: who conducted the interventions reported in the literature? The authors indicate that "Nurses have taken part in the interventions". It would be useful to complete this information with some data on who financed, designed and implemented the resilience promotion interventions. This may be useful to increase the practical use of the study results.

Author Response

Thank you so much for your dedication in reviewing this article. Your contributions and recommendations have been very useful in the approach to this research. Below, we will answer your suggestions point by point. Please see the attachment.

Abstract. According to editorial rules, it must be a single paragraph of about 200 words maximum, without headings.

As for the abstract, it has been structured according to the reviewer's recommendations: in a single paragraph and with less than 200 words (line 13-31).

The synthetic concepts reported after the abstract (“What does this paper contribute to the wider global clinical community?”) must be developed in the Discussion and reported in the Conclusions

The section "What does this paper contribute to the wider global clinical community?" has been removed as it is developed in the discussion and in the conclusions.

The interventions that the authors found are very heterogeneous, as they correctly recognize. An aspect that deserves to be underlined is that the experiences refer to very small samples. This inevitably limits the reliability of the results. Authors may report this issue to readers.

In the limitations section, it has been specified that, although four of the articles included in the scoping review are of mixed or qualitative design (prioritizing individual experiences over analytical metrics), the sample of participants has been small (line 274-276).

An element of interest to readers could be: who conducted the interventions reported in the literature? The authors indicate that "Nurses have taken part in the interventions". It would be useful to complete this information with some data on who financed, designed and implemented the resilience promotion interventions. This may be useful to increase the practical use of the study results.

In the scoping review, we did not include more professionals as one of the specific objectives was “To find out the implication of nursing professionals in the development of the intervention”.

At the request of the reviewer and considering that some articles do not indicated which professionals participated in the intervention, or it is very ambiguous with the use of terms such as "healthcare professionals", it is observed that many professionals who have carried out the interventions together with nurses, have been occupational therapists, and psychologists. In order not to duplicate this information, it was written in the data collection table and the discussion (line 241-242).

Reviewer 2 Report

Comments and Suggestions for Authors

First of all, I commend the authors for this very interesting article, but there are some issues that need to be addressed and justified in this study.

Methodology

It would be important to clarify the Alzheimer's disease severity of the subjects in the selected studies.

Provide justification for using CASP criteria for qualitative research and reference this tool.

Search Strategy

The temporal limit for the scope review is not clear.

In terms of databases, it would have been crucial to include Web of Science or Scopus, which are also highly relevant and may contain articles on this topic.

The PRISMA diagram should indicate the total number of articles found in each of the databases mentioned in the search strategy.

Discussion

It would be important to emphasize whether nursing interventions are carried out by other healthcare professionals in the articles selected in the scope review.

Conclusions:

It is important to expand this section by indicating what this research adds to the existing evidence. It would be interesting to discuss if there have been studies focused on adult populations without Alzheimer's disease but with other pathologies.

Finally I recommend to Include a section on limitations.

Author Response

Thank you so much for your dedication in reviewing this article. Your contributions and recommendations have been very useful in the approach to this research. Below, we will answer your suggestions, point by point.

Methodology

It would be important to clarify the Alzheimer's disease severity of the subjects in the selected studies.

In terms of the severity of Alzheimer's disease, all family caregivers of people with Alzheimer's disease were included in our scoping review, regardless of the level of disease severity. However, in view of the reviewer's recommendations, we have included in the data collection table the level of severity of Alzheimer's disease in three categories: early, middle, and final stage.

Provide justification for using CASP criteria for qualitative research and reference this tool

As shown in bibliographic reference number 29 (line 170-171), we have used the CaSP tool for analytical and qualitative studies. As for the latter, the CaSP reading tool is a checklist made up of 10 questions whose objective is to provide information standards for qualitative studies and evaluate the synthesis of qualitative evidence. We can provide a reference to this, but we consider that it is already referenced (line170). If you considered it necessary, we could add the following:

Cortés Rodríguez AE. Research methodology: from reader to disseminator. In: Editorial Universidad de Almería, editor. Research methodology: from reader to disseminator [Internet]. 1st ed. Almería; 2021 [cited 2024 Mar 25]. p. 126–32. Available from: https://repositori.uji.es/xmlui/bitstream/handle/10234/198391/78057.pdf?sequence=1&isAllowed=y

Search Strategy

The temporal limit for the scope review is not clear.

On line 126, the time limit for searching the databases for articles has been specified.

In terms of databases, it would have been crucial to include Web of Science or Scopus, which are also highly relevant and may contain articles on this topic.

Sorry for the confusion. The digital library from which we accessed the Medline database was through the Web of Science web portal, so both were treated as one. Fixed (line 119).

The PRISMA diagram should indicate the total number of articles found in each of the databases mentioned in the search strategy.

In accordance with the reviewer recommendation, the PRISMA flowchart has been modified and the total number of articles found in each of the databases mentioned in the search strategy has been indicated

It would be important to emphasize whether nursing interventions are carried out by other healthcare professionals in the articles selected in the scope review.

In the scoping review, we did not include more professionals as one of the specific objectives was “To find out the implication of nursing professionals in the development of the intervention”.

At the request of the reviewer and considering that some articles do not state which professionals participated in the intervention or are very ambiguous with the use of terms such as "healthcare professionals", it is observed that many professionals who have carried out the interventions have been occupational therapists, and psychologists. In order not to duplicate this information, it was written in the data collection table and the discussion (line 241-242).

It is important to expand this section by indicating what this research adds to the existing evidence. It would be interesting to discuss if there have been studies focused on adult populations without Alzheimer's disease but with other pathologies.

In the conclusion, a small section has been added to explain what the research adds, without duplicating the information written in the previous lines (line 295-298).

As for the recommendation to indicate other types of populations with other pathologies in which resilience is fostered, it is written in the introduction (line 87-90)

Finally I recommend to Include a section on limitations.

There is a section on limitations, before the conclusions (line 267-282)
